# Clinical Outcomes Associated with Parenteral Nutrition Caloric Provision in Geriatric Patients with Infectious Colitis [note 1]

**DOI:** 10.3390/nu17233707

**Published:** 2025-11-26

**Authors:** Yuro Kang, Jung Hwan Lee, Soo-Hyun Park, Somi Park, So-Youn Park, Tai Shun Yen, Jeongmi Park, Kye-Sook Kwon

**Affiliations:** 1College of Medicine, Inha University, Incheon 22212, Republic of Korea; yuro.kang@gmail.com; 2Department of Hospital Medicine, Inha University School of Medicine, Inha University Hospital, Incheon 22332, Republic of Korea; 3Department of Neurology, Soonchunhyang University Hospital Seoul, Seoul 04401, Republic of Korea; 4Department of Internal Medicine, Inha University School of Medicine, Inha University Hospital, Incheon 22332, Republic of Korea

**Keywords:** parenteral nutrition, geriatric patients, infectious colitis, caloric provision, hospitalization outcomes

## Abstract

**Background/Objectives:** Malnutrition is a critical determinant of adverse outcomes in geriatric patients. Infectious colitis, which often requires hospitalization due to dehydration and poor oral intake, frequently necessitates parenteral nutrition (PN) when enteral feeding is not feasible. However, evidence regarding the optimal caloric threshold for PN in geriatric patients with infectious colitis remains limited. **Methods:** In this single-center retrospective observational study, we analyzed 278 geriatric patients (≥65 years) admitted with infectious colitis to a tertiary hospital between 2008 and 2018. Patients hospitalized for more than 7 days or with alternative colitis etiologies were excluded. Daily PN caloric intake was categorized as >1000 kcal/day or ≤1000 kcal/day. Clinical and laboratory data were collected, and logistic regression was used to identify risk factors for prolonged hospitalization (>4 days). **Results:** Patients receiving > 1000 kcal/day PN had a significantly shorter mean length of stay compared with those receiving ≤ 1000 kcal/day (3.8 ± 1.5 vs. 4.2 ± 1.8 days, *p* = 0.03). In multivariable analysis, inadequate caloric intake (OR 1.90, 95% CI 1.08–3.34, *p* = 0.03), admission from a long-term care facility (OR 5.65, 95% CI 1.11–28.87, *p* = 0.04), elevated ESR (OR 1.74, 95% CI 1.03–2.94, *p* = 0.04), and lymphopenia (OR 1.95, 95% CI 1.06–3.61, *p* = 0.03) were independently associated with prolonged hospitalization. No significant difference in in-hospital mortality was observed between groups. **Conclusions:** Adequate PN supplementation exceeding 1000 kcal/day was associated with a shorter hospital stay in geriatric patients with infectious colitis.

## 1. Introduction

Parenteral nutrition (PN) is a vital intervention for patients unable to maintain adequate oral or enteral intake, providing essential macronutrients, electrolytes, trace elements, and vitamins intravenously [1,2]. As the global population ages, the number of hospitalized older adults continues to rise. Malnutrition among these patients is a major determinant of adverse outcomes, including prolonged length of stay (LOS) and increased mortality [3]. The high prevalence of frailty, anorexia, and reduced intake underscores the clinical need for evidence-based nutritional strategies in this population.

Infectious colitis is an acute inflammatory disorder of the colon caused by pathogenic microorganisms, most commonly bacteria [4]. The diagnosis is established when enteric pathogens are identified in stool cultures or when clinical findings necessitate targeted antibacterial therapy. Older adults with infectious colitis often require hospitalization due to dehydration and poor oral intake, and PN is frequently initiated when enteral feeding is not feasible [1,5,6,7].

Previous studies have shown that supplemental PN can shorten the LOS in critically ill patients [8]. However, limited evidence exists for geriatric patients with infectious colitis, who are particularly susceptible to malnutrition and delayed recovery. Moreover, no established consensus defines the optimal caloric threshold for PN provision in this population. Considering the altered metabolism associated with aging, increased energy demands during infection, and adverse effects of inadequate caloric supplementation, identifying an optimal PN caloric target in this group has substantial clinical importance.

In this study, we aimed to determine whether adequate parenteral caloric provision is associated with improved hospitalization outcomes in geriatric patients with infectious colitis and to identify an evidence-based caloric threshold to inform clinical nutritional practice.

## 2. Materials and Methods

### 2.1. Study Population

This single-center retrospective observational study was conducted at Inha University Hospital (Incheon, Korea) and included geriatric patients (≥65 years) diagnosed with infectious colitis who received total parenteral nutrition (TPN) and were hospitalized for ≤7 days between January 2008 and April 2018. Eligible patients were identified, and their clinical data were retrieved from the Electronic Medical Record (EMR) system. The 7-day exclusion in this study was designed to focus specifically on the acute phase of illness and to minimize confounding variables associated with prolonged hospitalization, such as severe complications or alternative etiologies [7]. The study was reviewed and approved by the ethics committee of Inha University Hospital (Approval No. 2025-10-018).

Infectious colitis was operationally defined as cases in which (1) pathogenic bacteria were detected in stool specimens through culture or polymerase chain reaction (PCR), or (2) antibacterial agents were administered under clinical suspicion of bacterial colitis [9,10].

Comprehensive demographic and clinical data were collected, including age, sex, body mass index (BMI), admission source, intensive care unit (ICU) admission requirement, presence of shock at presentation, and comorbid conditions assessed using the Charlson Comorbidity Index (CCI) [11,12]. Laboratory data obtained at admission included complete blood count parameters (white blood cell count, hemoglobin, platelet count), erythrocyte sedimentation rate (ESR), lymphocyte count, and biochemical values (albumin, cholesterol). Daily caloric intake from PN was calculated and divided into two categories: high-calorie (>1000 kcal/day) and low-calorie (≤1000 kcal/day) [13,14,15,16]. This threshold was selected as a pragmatic, post-hoc cutoff for this exploratory analysis, based on its utility in dividing the cohort for statistical comparison and informed by mean intakes reported in other previous observational studies. LOS, ICU stay, and in-hospital mortality were also recorded. LOS was dichotomized into >4 days and ≤4 days [17,18,19,20,21].

Given the acute illness and clinical high nutritional risk in this geriatric population, most patients initiated PN within 3 days of hospitalization. PN for geriatric patients with infection colitis included commercially available TPN and dextrose solutions. TPN was defined as the intravenous administration of a complete nutrient solution providing all essential macronutrients (carbohydrates, amino acids, and lipids), and water required to meet daily metabolic needs [2,22,23]. Dextrose solution refers to the D-isomer of glucose, a simple monosaccharide that serves as a primary energy source in PN or clinical settings (e.g., 5% or 10% dextrose solutions) [24,25]. Daily protein intake was calculated based on the standard amino acid composition of the specific commercial parenteral nutrition products administered, including MG TNA^®^ (MG Co., Ltd., Seoul, Republic of Korea), Winuf^®^ (JW Life Science, Seoul, Republic of Korea), SmofKabiven^®^ (Fresenius Kabi, Bad Homburg, Germany), and NuTRIflex^®^ (B. Braun Melsungen AG, Melsungen, Germany).

This article is a revised and expanded version of a conference abstract entitled ‘Parenteral Nutrition for Infectious Colitis in Geriatric Patients’, which was presented at the KSPEN (Korean Society for Parenteral and Enteral Nutrition) Annual Conference, Seongnam, Republic of Korea, 18 June 2019 [26].

### 2.2. Statistical Analysis

Baseline characteristics between the two caloric groups were compared using appropriate statistical tests. Categorical variables are expressed as frequencies and percentages, whereas continuous variables are presented as means and standard deviations (SD). Univariate and multivariable logistic regression analyses were performed to identify factors associated with prolonged hospitalization (>4 days). Multivariable logistic regression was specifically employed to adjust for potential confounding variables. Regarding missing data, data quality assessment revealed that the proportion of missing values for covariates included in the multivariable model was negligible (<5%). Given this low rate of missingness, a complete-case analysis approach was applied, excluding only the few cases with incomplete records to maintain data integrity without introducing significant bias. Univariate analyses were conducted using Chi-squared or Fisher’s exact tests for categorical variables. Results are presented as odds ratios (OR) with 95% confidence intervals (CI). Statistical significance was defined as *p* < 0.05. All analyses were performed using SPSS version 21.0 (SPSS Inc., Chicago, IL, USA).

## 3. Results

Figure 1 illustrates the patient selection process. A total of 1898 patients diagnosed with colitis and admitted via the emergency department were initially identified from the hospital database during the study period. Patients were systematically excluded based on diagnosis (399 patients with ischemic colitis, pseudomembranous colitis, inflammatory bowel disease, cytomegalovirus colitis, radiation colitis, viral enteritis, and tuberculosis enteritis), age (1061 patients younger than 65 years), and LOS (160 patients with LOS longer than 7 days). After applying all exclusion criteria, 278 geriatric patients (≥65 years) with suspected infectious bacterial enteritis who were hospitalized for 7 days or less were included in the final analysis.

As shown in Table 1, baseline characteristics were comparable between groups, except for a trend toward lower caloric supplementation among patients with chronic kidney disease (*p* = 0.05). Hemoglobin levels were slightly lower in the low-calorie group (12.5 ± 2.2 g/dL) than in the high-calorie group (13.1 ± 1.9 g/dL; *p* = 0.02). The high-calorie group also had a shorter mean hospital stay compared with the low-calorie group (3.8 ± 1.5 vs. 4.2 ± 1.8 days, *p* = 0.03). No significant difference in in-hospital mortality was observed between the two groups (1.1% vs. 2.2%, *p* = 0.53).

As expected, the mean caloric intake was significantly higher in the >1000 kcal/day group than in the ≤1000 kcal/day group (1748.0 ± 477.0 vs. 316.0 ± 293.0 kcal/day, *p* < 0.001). When normalized for body weight, the high-calorie group received a mean of 31.0 ± 10.3 kcal/kg/day and 0.71 ± 0.41 g/kg/day of protein. Conversely, the low-calorie group received substantially lower nutritional support, with a mean of 5.7 ± 5.4 kcal/kg/day and 0.05 ± 0.11 g/kg/day of protein (*p* < 0.001 for both). The distribution of PN type also differed significantly (*p* < 0.001), with all patients in the higher-calorie group receiving combined TPN with dextrose, whereas a substantial proportion of those in the low-calorie group received dextrose alone.

### Analysis of Risk Factors for Prolonged Hospitalization

In Table 2, the univariable analysis showed that lower BMI (≤25 kg/m^2^) was significantly associated with a longer LOS (*p* < 0.05), Nutritional factors also showed strong associations: patients with lower daily caloric intake (*p* < 0.01) and those receiving dextrose-only PN instead of combined TPN (*p* < 0.01) were more likely to remain hospitalized for more than 4 days.

In multivariate analysis, admission from a long-term care hospital (OR: 5.65, 95% CI, 1.11–28.87, *p* = 0.04), ESR greater than 22 mm/h (OR: 1.74, 95% CI, 1.03–2.94, *p* = 0.04), lymphocyte count less than 1500/mm^3^ (OR: 1.95, 95% CI, 1.06–3.61, *p* = 0.03) and lower PN caloric intake (OR: 1.90, 95% Cl, 1.08–3.34, *p* = 0.03) were significantly associated with hospitalization longer than 4 days. No significant differences were observed in ICU admission rates, presence of shock, or in-hospital mortality between groups stratified by daily PN caloric intake. The overall in-hospital mortality was low, with one death (1.1%) in the high-calorie group and four deaths (2.2%) in the low-calorie group (*p* = 0.53).

To verify the robustness of our results, we performed a sensitivity analysis excluding 39 patients admitted to the ICU. A multivariable logistic regression model was constructed using variables identified as significant (*p* < 0.1) in the univariable analysis of this non-ICU cohort. The analysis confirmed that BMI ≤ 25 kg/m^2^ (OR 1.95, 95% CI 1.04–3.67, *p* = 0.038), ESR > 22 mm/h (OR 1.94, 95% CI 1.08–3.47, *p* = 0.026), and admission from a long-term care hospital (OR 10.77, 95% CI 1.24–93.26, *p* = 0.031) remained statistically significant predictors. Low caloric intake (≤1000 kcal/day) also maintained a strong association with prolonged hospitalization (OR 1.82, 95% CI 0.99–3.33, *p* = 0.054), reinforcing the validity of the primary findings (Appendix A).

## 4. Discussion

This study demonstrates that supplementation with PN exceeding 1000 kcal per day was associated with shorter hospital stays in geriatric patients with infectious colitis. The key finding was that inadequate caloric intake (≤1000 kcal/day) was an independent risk factor for prolonged hospitalization, along with admission from a long-term care facility, elevated ESR, and lymphopenia. These results suggest that sufficient caloric support may facilitate recovery and reduce LOS, even in the absence of differences in mortality or ICU admission. Although caloric supplementation did not affect short-term mortality, its potential benefit in accelerating clinical recovery underscores the importance of optimal nutritional management in geriatric patients with infectious colitis.

Previous studies have shown that inadequate nutritional intake in older adults is closely associated with institutional discharge [16,27,28], elevated inflammatory markers [29,30] and lymphopenia [31,32]. In geriatric patients, underlying diseases and systemic inflammation increase basal metabolic rate, while reduced intake promotes a negative energy balance, underscoring the need for supplemental nutrition. Adequate caloric provision not only attenuates protein catabolism and weight loss but also helps preserve lymphocyte counts, modulate cytokine responses, and reduce infectious complications [33,34]. Moreover, sufficient nutritional support is essential for epithelial repair, wound healing, and organ function [35], while limiting muscle wasting facilitates earlier mobilization and rehabilitation. These mechanisms provide a plausible rationale for our findings, highlighting the importance of adequate PN in geriatric patients with infectious colitis. Our inclusion of dextrose solutions in our caloric analysis reflects the clinical reality of managing acute infectious colitis. According to clinical guidelines, the primary treatment goal involves rehydration to correct fluid and electrolyte deficits caused by diarrhea [7]. In this context, dextrose-containing fluids not only facilitate hydration but also serve as a supplementary energy source. Although dextrose alone does not meet full nutritional requirements, it provides essential carbohydrate calories that help spare endogenous protein catabolism and support metabolic function during periods of inadequate oral intake [36].

Regarding the timing of nutritional support, although American Society for Parenteral and Enteral Nutrition (ASPEN) guidelines recommend delaying PN for 5–7 days in well-nourished patients, they advocate for early initiation (as soon as feasible) in patients with moderate to severe malnutrition or those at high nutritional risk [37]. Our study population predominantly consisted of high-risk geriatric patients, as evidenced by poor nutritional and inflammatory markers, including BMI, hemoglobin, albumin, lymphocyte count, and elevated ESR. Furthermore, a significant proportion of patients were admitted from long-term care facilities, suggesting a baseline state of frailty and chronic malnutrition. Therefore, the early initiation of PN in this cohort was clinically justified to prevent further nutritional deterioration during the acute phase of infection.

Our findings are also consistent with international nutritional standards for geriatric and acutely ill populations. The European Society for Clinical Nutrition and Metabolism (ESPEN) guidelines recommend an energy intake of 27–30 kcal/kg/day for older adults to maintain nutritional status and support recovery [16]. Similarly, the ASPEN guidelines suggest a target of 25–30 kcal/kg/day for patients at high nutritional risk [38]. In our study, patients in the high-calorie group (>1000 kcal/day) received a mean of 31.0 ± 10.3 kcal/kg/day, effectively meeting or slightly exceeding these guideline targets. Conversely, the low-calorie group received only 5.7 ± 5.4 kcal/kg/day, representing a state of severe caloric deficit. This stark contrast highlights that the 1000 kcal/day threshold served as a practical surrogate for guideline-concordant nutritional support in this geriatric cohort. The shorter length of stay observed in the high-calorie group supports the clinical benefit of meeting these established energy requirements during the acute phase of infectious colitis.

This study had several notable strengths. First, it specifically focused on geriatric patients with infectious colitis–a vulnerable population in whom nutritional management is often overlooked–and identified a practical caloric cutoff of 1000 kcal/day for PN provision. Unlike previous studies that primarily examined critically ill patients or broadly addressed malnutrition in older adults, our study provides disease-specific evidence applicable to clinical practice. Second, by conducting a multivariable analysis, we confirmed that inadequate caloric intake was an independent risk factor for prolonged hospitalization, along with admission from long-term care facilities, elevated ESR, and lymphopenia. This comprehensive risk assessment highlights the multifactorial determinants of clinical outcomes and emphasizes the importance of sufficient caloric support in promoting recovery.

However, this study also had some limitations. First, the retrospective, single-center design inevitably limits the ability to establish definitive causal relationships. There is a potential risk of reverse causality, where greater disease severity may have led to reduced caloric intake rather than the reverse. However, we attempted to mitigate this bias by adjusting for key severity markers (e.g., ICU admission, shock, CCI) in multivariable models and by confirming the robustness of our findings through a sensitivity analysis excluding critically ill patients. Second, because data were obtained exclusively from a tertiary academic hospital, the generalizability of the findings to other healthcare settings or populations may be limited. Third, the study’s sample size was modest, partly due to the strict inclusion criterion of hospitalization for 7 days or less. While this threshold was strategically chosen to minimize heterogeneity and focus on the acute phase of infectious colitis, it may have reduced the statistical power to detect subtle effects for secondary outcomes, thereby increasing the risk of Type II errors.

Fourth, formal nutritional assessment tools such as the Nutrition Risk Screening 2002 (NRS-2002) or the Global Leadership Initiative on Malnutrition (GLIM) criteria were not consistently available due to the retrospective nature of the study [39,40]. However, malnutrition status was evaluated using clinical indicators (e.g., poor oral intake, dehydration, frailty) and laboratory markers mentioned above, which served as surrogate measures to guide the initiation of parenteral nutrition. In addition, reliance on electronic medical records limited the availability and granularity of some key data, such as oral intake assessment, nutritional composition, and inter-provider variability in PN administration. Important confounding variables—such as pre-existing nutritional status, detailed comorbidity burden, and functional outcomes—were not fully captured. Furthermore, the reproducibility of these findings in other healthcare settings remains to be established due to the single-center design. Additionally, by focusing exclusively on short-term hospitalization outcomes, the study was unable to assess longer-term endpoints such as readmission, post-discharge mortality, or nutritional recovery. These limitations highlight the need for future multicenter, prospective studies with larger sample sizes, extended follow-up, and more comprehensive evaluation of potential confounders to validate and expand upon these findings.

## 5. Conclusions

This study provides novel evidence that adequate PN supplementation exceeding 1000 kcal/day was associated with faster recovery and shorter hospitalization in geriatric patients with infectious colitis. By establishing a practical caloric threshold, these findings offer clinicians a concrete reference for guiding nutritional support in this vulnerable population. Proactive application of PN strategies based on these results has the potential to enhance patient recovery, optimize hospital resource utilization, and ultimately improve the quality of care for older adults with infectious colitis. Incorporating these findings into clinical guidelines may help advance personalized nutrition management and strengthen outcomes in geriatric medicine.

## Figures and Tables

**Figure 1 nutrients-17-03707-f001:**
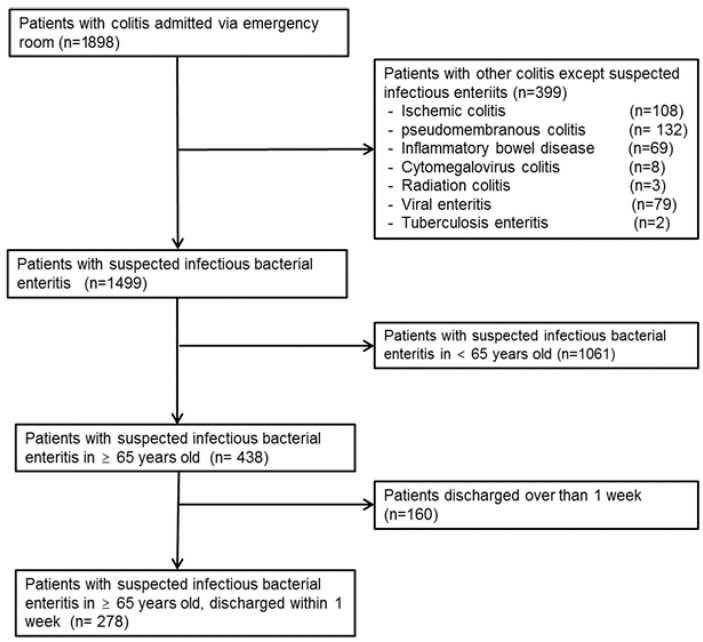
Flow chart of patients’ selection.

**Table 1 nutrients-17-03707-t001:** Baseline characteristics according to parenteral nutrition calories per day.

Characteristics, *n* (%)	≤1000 kcal/day (*n* = 186)	>1000 kcal/day (*n* = 92)	*p*-Value
Age, year, mean ± SD	74.4 ± 6.5	75.7 ± 7.1	0.1
Sex			0.74
Male	61 (32.8)	32	
Female	125 (67.2)	60	
BMI kg/m^2^, mean ± SD	23.7 ± 3.6	23.7 ± 3.6	0.97
Long-term care facility	8 (4.3)	1 (1.1)	0.15
Intensive care unit	31 (16.7)	8 (8.7)	0.07
Shock	25 (13.4)	6 (6.5)	0.08
Co-existing morbidity			
Diabetes			0.83
Uncomplicated	60 (32.3)	33 (35.9)	
Complicated	6 (3.2)	3 (3.3)	
Liver disease	13 (7)	1 (1.1)	0.105
Malignancy			0.539
Any leukemia, lymphoma, or localized solid tumor	9 (4.8)	2 (2.2)	
Metastatic solid tumor	5 (2.7)	2 (2.2)	
Chronic kidney disease	12 (6.5)	1 (1.1)	0.05
Congestive heart failure	21 (11.3)	17 (18.5)	0.101
Peripheral artery occlusive disease	2 (1.1)	3 (3.3)	0.2
Chronic obstructive pulmonary disease	4 (2.2)	3 (3.3)	0.58
Cerebral vascular attack	26 (14.0)	14 (15.2)	0.78
Hemiplegia	2 (1.1)	0 (0)	0.31
Rheumatic disease	5 (2.7)	2 (2.2)	0.8
Dementia	11 (5.9)	3 (3.3)	0.34
Peptic ulcer	2 (1.1)	3 (3.3)	0.2
Charlson comorbidity index	1.1 ± 1.5	1.0 ± 1.2	0.70
Laboratory exam			
White blood cell, ×1000/μL, mean ± SD	12.7 ± 22.6	10.9 ± 6.2	0.46
Hemoglobin, g/dL, mean ± SD	12.5 ± 2.2	13.1 ± 1.9	0.02
Platelet, ×1000/μL, mean ± SD	228.7 ± 102.0	212.7 ± 69.6	0.18
ESR mm/h, mean ± SD	32.6 ± 26.2	29.6 ± 24.3	0.36
Lymphocyte count, /μL, mean ± SD	1280.2 ± 1395.5	1249.2 ± 1004.1	0.85
Albumin, g/dL, mean ± SD	4.5 ± 10.5.	3.9 ± 1.0	0.6
Cholesterol, mg/dL, mean ± SD	143.9 ± 51.8	138.4 ± 47.5	0.4
Nutrition			
Kcal/day	316.0 ± 293.0	1748.0 ± 477.0	<0.001
Caloric intake, kcal/kg/day, mean ± SD	5.7 ± 5.4	31.0 ± 10.3	<0.001
Protein intake, g/kg/day, mean ± SD	0.05 ± 0.11	0.71 ± 0.41	<0.001
Parenteral nutrition category			<0.001
Only dextrose	71 (38.2)	0 (0)	
Combined TPN with dextrose fluid	77 (41.4)	92 (100)	
Outcomes			
Length of stay in hospital, day, mean ± SD	4.2 ± 1.8	3.8 ± 1.5	0.03
In-hospital mortality	4 (2.2)	1 (1.1)	0.53

BMI: body mass index, ESR: erythrocyte sedimentation rate, TPN: total parenteral nutrition.

**Table 2 nutrients-17-03707-t002:** Multivariable analysis for longer than 4 days hospital duration.

				Univariable Analysis			Multivariable Analysis		
		Total	>4 day	Odd Ratio	95% CI	*p*-Value	Odd Ratio	95% CI	*p*-Value
Age	≤75 years	156	62 (39.7)	1		0.53			
	>75 years	122	44 (36.1)	0.86	(0.53–1.40)				
Sex	Male	93	32 (34.4)	1		0.36			
	female	185	74 (40.0)	1.24	(0.74–2.08)				
Body mass index	>25 kg/m^2^	76	24 (31.6)	1		0.023	1		0.056
	≤25 kg/m^2^	202	82 (40.6)	1.86	(1.09–3.17)		1.74	(0.99–3.08)	
Long-term care hospital	No	269	99 (36.8)	1		0.013	1		0.04
	Yes	9	7 (77.8)	5.282	(1.19–28.57)		5.65	(1.11–28.87)	
Intensive care unit	No	239	83 (34.7)	1		0.004	1		0.06
	Yes	39	23 (59)	2.60	(1.31–5.20)		2.2	(0.98–4.93)	
Diabetes	No	176	65 (36.9)	1		0.59			
	Yes	102	41 (40.2)	1.21	(0.73–2.02)				
Liver disease	No	264	101 (38.3)	1		0.85			
	Yes	14	5 (35.7)	0.98	(0.31–3.09)				
Malignancy	No	260	97 (37.3)	1		0.28			
	Yes	18	9 (50.0)	1.63	(0.62–4.24)				
Chronic kidney disease	No	265	100 (37.7)	1		0.54			
	Yes	13	6 (46.2)	1.37	(0.45–4.19)				
Congestive heart failure	No	240	96 (40.0)	1		0.11			
	Yes	38	10 (26.3)	0.52	(0.24–1.11)				
PAOD	No	273	106 (38.8)	1		0.99			
	Yes	5	0 (0.0)	-	-				
COPD	No	271	102 (37.6)	1		0.3			
	Yes	7	4 (57.1)	2.14	(0.47–9.76)				
Cerebral vascular attack	No	238	87 (36.6)	1		0.19			
	Yes	40	19 (47.5)	1.51	(0.47–9.76)				
Hemiplegia	No	276	105 (38.0)	1		0.73			
	Yes	2	1 (50)	1.58	(0.10–25.52)				
Rheumatic disease	No	271	103 (38)	1		0.79			
	Yes	7	3 (42.9)	1.19	(0.26–5.40)				
Dementia	No	264	99 (37.5)	1		0.35			
	Yes	14	7 (50)	1.61	(0.55–4.74)				
Peptic ulcer	No	273	106 (38.8)	1		0.99			
	Yes	5	0 (0)	-	-				
Charlson comorbidity index	≤3	262	98 (37.4)	1		0.31			
	>3	16	8 (50)	1.62	(0.59–4.45)				
Shock	No	247	89 (36.0)	1		0.04	1		0.6
	Yes	31	17 (54.8)	2.08	(1.02–4.42)		1.27	(0.52–3.11)	
White blood cell, /mm^3^	4000–10,000	126	43 (34.1)	1		0.21			
	>10,000 or ≤4000	152	63 (41.4)	1.44	(0.88–2.35)				
Hemoglobin, g/dL	>12	185	70 (37.8)	1		0.88			
	≤12	93	36 (38.7)	0.95	(0.57–1.56)				
Platelet count, ×1000/mm^3^	>15 K	229	87 (38)	1		0.91			
	≤15 K	49	19 (38.8)	1.00	(0.53–1.88)				
ESR, mm/h	≤22	115	35 (30.4)	1		0.03	1		0.04
	>22	163	71 (43.6)	1.74	(1.05–2.87)		1.74	(1.03–2.94)	
Lymphocyte count, /mm^3^	>1500	74	20 (27.0)	1		0.02	1		0.03
	≤1500	204	86 (42.2)	2.05	(1.14–3.67)		1.95	(1.06–3.61)	
Albumin, g/dL	>3.3	228	86 (37.7)	1		0.76			
	≤3.3	50	20 (40.0)	1.67	(0.95–2.94)				
Calories per day, kcal/day	>1000	186	81 (43.5)	1		0.002	1		0.03
	≤1000	92	25 (27.2)	1.86	(1.09–3.17)		1.9	(1.08–3.34)	

PAOD: peripheral arterial occlusive disease, COPD: chronic obstructive pulmonary disease, ESR: erythrocyte sedimentation rate.

## Data Availability

The data presented in this study are available upon request from the corresponding authors. The data are not publicly available due to privacy and ethical restrictions.

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
