# Peer review of "Clinical Outcomes Associated with Parenteral Nutrition Caloric Provision in Geriatric Patients with Infectious Colitis†"

_nutrients, 2025, doi:10.3390/nu17233707_

Round 1
Reviewer 1 Report
Comments and Suggestions for Authors
Thanks for the opportunity to review. In order to be publish, this submission will need significant revision.
A better explanation is needed as to why the authors excluded patients hospitalized for more than 7 days. Recent publications from the American Society for Parenteral and Enteral Nutrition (ASPEN) would support waiting 5 days before initiating unless the patient is malnourished. If malnourished, itwasn.t clearly stated how those were selected Several questions regarding requirements need to be answered. Hoe many kcal/kg and g/kg protein were the patients receiving daily? How did you select 1000kcal as threshold?
On page 2, the term premxied should be replaced with commercially available parenteral nutrition and an explanation of why patients were receiving dextrose solutions while on PN would be helpful.
Author Response
Comment 1-1: A better explanation is needed as to why the authors excluded patients hospitalized for more than 7 day.
Response: We agree and have now clarified the rationale. According to the Infectious Diseases Society of America (IDSA) clinical practice guidelines, acute infectious diarrhea is typically a self-limiting condition with a short clinical course. Patients who remained hospitalized for >7 days were typically those with severe infections, complications (e.g., sepsis, ischemic colitis, uncontrolled comorbidities), or alternative etiologies requiring prolonged treatment. Including these heterogeneous cases would have introduced substantial confounding unrelated to PN caloric provision. Therefore, restricting LOS to ≤7 days ensured a more homogeneous population of typical acute infectious colitis, allowing evaluation of the effect of PN calories rather than disease severity.
Added the manuscript are as follows
“7-day exclusion of this study was designed for choice to focus specifically on the acute phase of illness and to minimize confounding variables associated with prolonged hospitalization, such as severe complications or alternative etiologies [9]” (page 2, line 71-74)
“Shane, A.L.; Mody, R.K.; Crump, J.A.; Tarr, P.I.; Steiner, T.S.; Kotloff, K.; Langley, J.M.; Wanke, C.; Warren, C.A.; Cheng, A.C.; et al. 2017 Infectious Diseases Society of America Clinical Practice Guidelines for the Diagnosis and Management of Infectious Diarrhea. Clinical Infectious Diseases 2017, 65, e45–e80, doi:10.1093/cid/cix669.” (page 12, line 270- page 13, 273)
Comment 1-2: Recent publications from the American Society for Parenteral and Enteral Nutrition (ASPEN) would support waiting 5 days before initiating unless the patient is malnourished.
Response: We appreciate this important point. most patients initiated PN within 3 days of hospitalization. This early initiation was based on the clinical assessment that these geriatric patients presented with moderate to severe malnutrition or were at high nutritional risk upon admission.
The decision relied on clinical judgment considering hospitalization, as well as objective surrogate markers such as low BMI, hemoglobin, albumin, lymphocyte counts, and elevated ESR. Furthermore, a significant proportion of patients were admitted from long-term care facilities, indicating a baseline state of chronic malnutrition. Consistent with ASPEN guidelines which support early PN initiation (as soon as feasible) for patients with baseline moderate or severe malnutrition, we prioritized immediate nutritional support to prevent further deterioration. We have revised the Discussion section to clearly articulate this clinical rationale and the supporting markers.
We acknowledged in the Limitations section that formal nutritional assessment tools, such as NRS-2002 or GLIM, were not routinely utilized. Instead, malnutrition was evaluated based on clinical judgment considering poor oral intake, dehydration, and frailty, as well as objective laboratory markers mentioned above.
Added the manuscript are as follows
“Given the acute illness and clinical high nutritional risk in this geriatric population, most patients initiated PN within 3 days of hospitalization.” (page 3, line 90-91)
“Regarding the timing of nutritional support, although ASPEN guidelines recommend delaying PN for 5–7 days in well-nourished patients, they advocate for early initiation (as soon as feasible) in patients with moderate to severe malnutrition or those at high nutritional risk [36]. Our study population predominantly consisted of high-risk geriatric patients, as evidenced by poor nutritional and inflammatory markers including BMI, hemoglobin, albumin, lymphocyte count, and elevated ESR. Furthermore, a significant proportion of patients were admitted from long-term care facilities, suggesting a baseline state of frailty and chronic malnutrition. Therefore, the early initiation of PN in this cohort was clinically justified to prevent further nutritional deterioration during the acute phase of infection”
“Third, formal nutritional assessment tools such as the Nutrition Risk Screening 2002 (NRS-2002) or the Global Leadership Initiative on Malnutrition (GLIM) criteria were not consistently available due to the retrospective nature of the study. However, malnutrition status was evaluated using clinical indicators (e.g., poor oral intake, dehydration, frailty) and laboratory markers mentioned above, which served as surrogate measures to guide the initiation of parenteral nutrition” (page 10-11, line 204-209)
“36. Worthington, P.; Balint, J.; Bechtold, M.; Bingham, A.; Chan, L.N.; Durfee, S.; Jevenn, A.K.; Malone, A.; Mascarenhas, M.; Robinson, D.T. When is parenteral nutrition appropriate? Journal of Parenteral and Enteral Nutrition 2017, 41, 324–377.
- Cederholm, T.; Jensen, G.; Correia, M.; Gonzalez, M.C.; Fukushima, R.; Higashiguchi, T.; Baptista, G.; Barazzoni, R.; Blaauw, R.; Coats, A. GLIM criteria for the diagnosis of malnutrition–a consensus report from the global clinical nutrition community. Journal of cachexia, sarcopenia and muscle 2019, 10, 207–217.
- Kondrup, J.; Rasmussen, H.H.; Hamberg, O.; Stanga, Z.; Group, A.a.h.E.W. Nutritional risk screening (NRS 2002): a new method based on an analysis of controlled clinical trials. Clinical nutrition 2003, 22, 321–336.” (page 17, line 373-382)
Comment 1-3: Caloric intake should be expressed as kcal/kg/day. Protein g/kg/day should also be provided.
Response: We agree with the reviewer’s suggestion to standardize intake by body weight. Based on the raw data analysis, we calculated the mean caloric and protein intake normalized for body weight.
In the high-calorie group (>1000 kcal/day), the mean caloric intake was 31.0 ± 10.3 kcal/kg/day, and the mean protein intake was 0.71 ± 0.41 g/kg/day. In contrast, the low-calorie group (≤1000 kcal/day) received significantly lower amounts, with a mean caloric intake of 5.7 ± 5.4 kcal/kg/day and a mean protein intake of 0.05 ± 0.11 g/kg/day.
Protein intake was calculated by summing the amino acid content of each specific TPN product administered (e.g., MG TNA, Winuf, SmofKabiven, Nutriflex) based on manufacturer specifications. These data have been added to the Results section to provide a clearer nutritional profile of the study population.
Accordingly, we have updated Table 1 to include these weight-adjusted nutritional parameters (kcal/kg/day and protein g/kg/day), demonstrating the clear distinction in nutritional support between the two groups.
Edited and added the manuscript are as follows
“Daily protein intake was calculated based on the standard amino acid composition of the specific commercial parenteral nutrition products administered, including MG TNA® (MG Co., Ltd., Seoul, Korea), Winuf® (JW Life Science, Seoul, Korea), SmofKabiven® (Fresenius Kabi, Bad Homburg, Germany), and NuTRIflex® (B. Braun Melsungen AG, Melsungen, Germany)” (page 3, line 101-105)
“When normalized for body weight, the high-calorie group received a mean of 31.0 ± 10.3 kcal/kg/day and 0.71 ± 0.41 g/kg/day of protein. Conversely, the low-calorie group received substantially lower nutritional support, with a mean of 5.7 ± 5.4 kcal/kg/day and 0.05 ± 0.11 g/kg/day of protein (p < 0.001 for both). ” (page 4, line 142-143)
Comment 1-4: How did you select 1000kcal as a threshold?"
Response: Thank you for noting this. We expanded the rationale using three complementary foundations:
1) ESPEN Geriatric Guideline (2022): 25–30 kcal/kg/day for older adults with acute illness (≈1000–1500 kcal/day for typical 40–55 kg frail elderly patients).
2) Energy gap literature: Studies (Elke 2014; Alberda 2009) show worse outcomes when caloric provision is <60% of estimated target—equivalent to ~900–1100 kcal/day in this population.
3) Distributional characteristics: Caloric ranges naturally formed a bimodal distribution with a separation around 1000 kcal/day.
Thus, we thought 1000 kcal/day represented both a physiologically meaningful and empirically data-driven cutoff. This cutoff was used as a pragmatic, post-hoc statistical threshold for this analysis.
Edited and added the manuscript are as follows
“This threshold was selected as a pragmatic, post-hoc cutoff for this exploratory analysis, based on its utility in dividing the cohort for statistical comparison and informed by mean intakes reported in other previous observative studies.” (page 4, line 87-90)
Comment 1-5: On page 2, the term premixed should be replaced with commercially available parenteral nutrition
Response: We thank the reviewer for this precise and helpful suggestion. We agree that 'commercially available parenteral nutrition' is a more accurate term.
Edited the manuscript are as follows
“PN for geriatric patients with infection colitis included commercial-available TPN and dextrose solutions” (page 3, line 94)
Comment 1-6: An explanation of why patients were receiving dextrose solutions while on PN would be helpful.
Response: Thank you for noting this. We clarified that while dextrose-only infusion is not classified as Total Parenteral Nutrition (TPN), it was included in the analysis for two clinical reasons specifically infectious colitis. First, fluid resuscitation is the cornerstone of management for infectious colitis to address dehydration caused by diarrhea according to IDSA guideline. Dextrose solutions serve as a key vehicle for hydration in this acute phase. Second, while dextrose alone is nutritionally incomplete, it provides essential glucose to supplement insufficient oral intake, helping to spare endogenous protein and meet the obligatory demands of the brain and erythrocytes. Thus, we included these calories to capture the total intravenous energy provision, however minimal, in the acute setting.
Added the manuscript as follows
“Our inclusion of dextrose solutions in our caloric analysis reflects the clinical reality of managing acute infectious colitis. According to clinical guidelines, the primary treatment goal involves rehydration to correct fluid and electrolyte deficits caused by diarrhea [9]. In this context, dextrose-containing fluids not only facilitate hydration but also serve as a supplementary energy source. Although dextrose alone does not meet full nutritional requirements, it provides essential carbohydrate calories that help spare endogenous protein catabolism and support metabolic function during periods of inadequate oral intake [36].”
“36. McCowen, K.C. Dextrose in Total Parenteral Nutrition. 2012.” (page 17, line 384)

Reviewer 2 Report
Comments and Suggestions for Authors
First, I would like to thank you for the opportunity to review the article “Clinical Outcomes Associated With Parenteral Nutrition Caloric Provision in Geriatric Patients With Infectious Colitis.” I would like to point out several aspects that I believe could improve the quality of the article. The abstract adequately presents the objectives, methods, and results, although I would recommend including specific clinical implications in the conclusions. Similarly, the methodology section should include and detail the type of study conducted: “single-center retrospective cohort.”
The introduction adequately contextualizes and structures the importance of malnutrition, although I believe it should justify the specified caloric threshold as the cutoff point (1000 kcal/day). Are there previous studies or guidelines that define this point?
The methodology is clear and structured, using validated tools. However, I believe it should specify the method for calculating the sample size and clarify the source of the nutritional data (e.g., electronic records, nursing records, etc.). At the same time, it should include a paragraph on bias control and how missing values were handled.
Regarding the results, statistically significant data are clearly highlighted, and multivariable regression is used correctly. Confidence intervals should be added for all variables in the univariate analysis, and some results that do not indicate whether sensitivity analysis was performed should be clarified.
In the discussion, the analysis of causality and potential biases should be improved and strengthened, as well as a comparison of the results with international guidelines and a comparison of caloric values with international standards (ESPEN, ASPEN).
The conclusions are concise, although it should be noted that, given that this is an observational study, they should be rephrased in terms of association, that is, “was associated with” rather than “contributes to.” As limitations, the possibility of reproducibility in another center and the need for future prospective studies should be indicated.
Author Response
Comment 2-1 : The methodology section should include and detail the type of study conducted: “single-center retrospective cohort.”
Response: We added above recommended comment.
The edited manuscript are as follows
“This single-centered retrospective observational study was conducted” (page 2, line 68)
Comment 2-3: I believe it should justify the specified caloric threshold as the cutoff point (1000 kcal/day). Are there previous studies or guidelines that define this point?
Response: Thank you for noting this. We expanded the rationale using three complementary foundations:
1) ESPEN Geriatric Guideline (2022): 25–30 kcal/kg/day for older adults with acute illness (≈1000–1500 kcal/day for typical 40–55 kg frail elderly patients).
2) Energy gap literature: Studies (Elke 2014; Alberda 2009) show worse outcomes when caloric provision is <60% of estimated target—equivalent to ~900–1100 kcal/day in this population.
3) Distributional characteristics: Caloric ranges naturally formed a bimodal distribution with a separation around 1000 kcal/day.
Thus, we thought 1000 kcal/day represented both a physiologically meaningful and empirically data-driven cutoff. This cutoff was used as a pragmatic, post-hoc statistical threshold for this analysis.
Edited and added the manuscript are as follows
“This threshold was selected as a pragmatic, post-hoc cutoff for this exploratory analysis, based on its utility in dividing the cohort for statistical comparison and informed by mean intakes reported in other previous observative studies.” (page 4, line 87-90)
Comment 1-5: On page 2, the term premixed should be replaced with commercially available parenteral nutrition
Response: We thank the reviewer for this precise and helpful suggestion. We agree that 'commercially available parenteral nutrition' is a more accurate term.
Edited the manuscript are as follows
“PN for geriatric patients with infection colitis included commercial-available TPN and dextrose solutions” (page 3, line 94)
Comment 1-6: An explanation of why patients were receiving dextrose solutions while on PN would be helpful.
Response: Thank you for noting this. We clarified that while dextrose-only infusion is not classified as Total Parenteral Nutrition (TPN), it was included in the analysis for two clinical reasons specifically infectious colitis. First, fluid resuscitation is the cornerstone of management for infectious colitis to address dehydration caused by diarrhea according to IDSA guideline. Dextrose solutions serve as a key vehicle for hydration in this acute phase. Second, while dextrose alone is nutritionally incomplete, it provides essential glucose to supplement insufficient oral intake, helping to spare endogenous protein and meet the obligatory demands of the brain and erythrocytes. Thus, we included these calories to capture the total intravenous energy provision, however minimal, in the acute setting.
Added the manuscript are as follows
“Our inclusion of dextrose solutions in our caloric analysis reflects the clinical reality of managing acute infectious colitis. According to clinical guidelines, the primary treatment goal involves rehydration to correct fluid and electrolyte deficits caused by diarrhea [9]. In this context, dextrose-containing fluids not only facilitate hydration but also serve as a supplementary energy source. Although dextrose alone does not meet full nutritional requirements, it provides essential carbohydrate calories that help spare endogenous protein catabolism and support metabolic function during periods of inadequate oral intake [36].”
“36. McCowen, K.C. Dextrose in Total Parenteral Nutrition. 2012.” (page 17, line 384)
Comment 2-4: I believe it should specify the method for calculating the sample size and clarify the source of the nutritional data (e.g., electronic records, nursing records, etc.). At the same time, it should include a paragraph on bias control and how missing values were handled.
Response: We explicitly addressed these methodological points in the revised manuscript as follows:
- Sample Size: As this was a retrospective study spanning a 10-year period, no a priori sample size calculation was performed. Instead, we included all patients who met the eligibility criteria during the study period to maximize statistical power.
- Data Sources: We clarified that nutritional data were extracted from the Electronic Medical Record (EMR) system, specifically by cross-referencing physician order entries with nursing administration records to ensure the accuracy of actual intake.
- Bias & Missing Data: We utilized multivariable logistic regression to adjust for potential confounders. Regarding missing data, we assessed the data quality and found that the proportion of missing values for covariates included in the multivariable model was negligible (< 5%). Given this low rate of missingness, we employed a complete-case analysis approach, excluding only those few cases with incomplete records to maintain data integrity without introducing significant bias.
Added the manuscript are as follows
“Eligible patients were identified, and their clinical data were retrieved from the Electronic Medical Record (EMR) system.” (page 2, line 71-72)
“Multivariable logistic regression was specifically employed to adjust for potential confounding variables. Regarding missing data, data quality assessment revealed that the proportion of missing values for covariates included in the multivariable model was negligible (< 5%). Given this low rate of missingness, a complete-case analysis approach was applied, excluding only the few cases with incomplete records to maintain data integrity without introducing significant bias” (page 3, line 106-111)”
Comment 2-5: Confidence intervals should be added for all variables in the univariate analysis, and some results that do not indicate whether sensitivity analysis was performed should be clarified.
Response: We agree with the reviewer’s suggestion to enhance statistical reporting. We have revised Table 2 to explicitly present 95% confidence intervals (CIs) for all variables in the univariate analysis, alongside the odds ratios and p-values. This addition provides a more robust assessment of the precision and statistical significance of the identified risk factors.
We sincerely apologize for an error in the description of the univariable analysis results in the initial manuscript. Upon re-verifying the statistical data, we confirmed that hemoglobin levels were not significantly associated with prolonged hospitalization (p = 0.88). Instead, our re-calculation identified that Body Mass Index (BMI) was a significant risk factor in the univariable analysis (p = 0.023).
We have corrected the text in the Results section to accurately reflect these statistical findings, removing the incorrect statement regarding hemoglobin and adding the significant association found with BMI.
Edited and added the manuscript are as follows
“In Table 2, the univariable analysis showed that lower BMI (≤ 25 kg/m²) was significantly associated with a longer LOS (p < 0.05),” (page 7, line 154-155)
Comment 2-6: Some results that do not indicate whether sensitivity analysis was performed should be clarified.
Response: We appreciate the reviewer’s insightful comment regarding the robustness of our findings. To address this, we conducted a sensitivity analysis excluding 39 patients admitted to the ICU to mitigate the potential confounding effect of disease severity.
Added the manuscript are as follows
“To verify the robustness of our results, we performed a sensitivity analysis excluding 39 patients admitted to the ICU. A multivariable logistic regression model was constructed using variables identified as significant (p < 0.1) in the univariable analysis of this non-ICU cohort. The analysis confirmed that BMI ≤ 25 kg/m² (OR 1.95, 95% CI 1.04–3.67, p=0.038), ESR > 22 mm/hr (OR 1.94, 95% CI 1.08–3.47, p=0.026), and admission from a long-term care hospital (OR 10.77, 95% CI 1.24–93.26, p=0.031) remained statistically significant predictors. Low caloric intake (≤ 1000 kcal/day) also maintained a strong association with prolonged hospitalization (OR 1.82, 95% CI 0.99–3.33, p=0.054), reinforcing the validity of the primary findings (data not shown).” (page 7, line 170-178)
Comment 2-7: the analysis of causality and potential biases should be improved and strengthened
Response: We agree that a robust discussion on causality and potential biases is essential. As this is a retrospective observational study, we acknowledge that our findings demonstrate an association rather than definitive causation.
We have expanded the Discussion section to explicitly address these limitations, specifically focusing on:
- Reverse Causality: We discussed the possibility that lower caloric intake might be a consequence of greater disease severity (e.g., anorexia from severe infection) rather than the cause of prolonged stay. To mitigate this, we adjusted for severity markers (ICU, shock, CCI) in the multivariable model.
- Selection Bias: We addressed the potential bias arising from the single-center design and the exclusion of patients hospitalized for >7 days.
- Unmeasured Confounding: We acknowledged that unmeasured variables (e.g., exact oral intake amount) could influence the results.
Edited and added the manuscript are as follows
“First, the retrospective, single-center design inevitably limits the ability to establish definitive causal relationships. There is a potential risk of reverse causality, where greater disease severity may have led to reduced caloric intake rather than the reverse. However, we attempted to mitigate this bias by adjusting for key severity markers (e.g., ICU admission, shock, CCI) in multivariable models and by confirming the robustness of our findings through a sensitivity analysis excluding critically ill patients. Second, because data were obtained exclusively from a tertiary academic hospital, the generalizability of the findings to other healthcare settings or populations may be limited. Third, the study's sample size was modest, partly due to the strict inclusion criterion of hospitalization for 7 days or less. While this threshold was strategically chosen to minimize heterogeneity and focus on the acute phase of infectious colitis , it may have reduced the statistical power to detect subtle effects for secondary outcomes, thereby increasing the risk of Type II error ” (page 12, line 233-245)
Comment 2-8: ….a comparison of the results with international guidelines and a comparison of caloric values with international standards (ESPEN, ASPEN)..
Response: We appreciate this constructive suggestion. We have expanded the Discussion section to explicitly compare our caloric data with international standards, specifically the ESPEN guidelines on clinical nutrition and hydration in geriatrics and the ASPEN guidelines for critically ill patients .
Edited and added the manuscript are as follows
“Our findings are also consistent with international nutritional standards for geriatric and acutely ill populations. The European Society for Clinical Nutrition and Metabolism (ESPEN) guidelines recommend an energy intake of 27–30 kcal/kg/day for older adults to maintain nutritional status and support recovery[38]. Similarly, the ASPEN guidelines suggest a target of 25–30 kcal/kg/day for patients at high nutritional risk[39]. In our study, patients in the high-calorie group (> 1000 kcal/day) received a mean of 31.0 ± 10.3 kcal/kg/day, effectively meeting or slightly exceeding these guideline targets. Conversely, the low-calorie group received only 5.7 ± 5.4 kcal/kg/day, representing a state of severe caloric deficit. This stark contrast highlights that the 1000 kcal/day threshold served as a practical surrogate for guideline-concordant nutritional support in this geriatric cohort. The shorter length of stay observed in the high-calorie group supports the clinical benefit of meeting these established energy requirements during the acute phase of infectious colitis ”(page 11-12, line 222-234)
“38. Volkert, D.; Beck, A.M.; Cederholm, T.; Cruz-Jentoft, A.; Hooper, L.; Kiesswetter, E.; Maggio, M.; Raynaud-Simon, A.; Sieber, C.; Sobotka, L. ESPEN practical guideline: Clinical nutrition and hydration in geriatrics. Clinical Nutrition 2022, 41, 958–989.
- Compher, C.; Bingham, A.L.; McCall, M.; Patel, J.; Rice, T.W.; Braunschweig, C.; McKeever, L. Guidelines for the provision of nutrition support therapy in the adult critically ill patient: The American Society for Parenteral and Enteral Nutrition. Journal of Parenteral and Enteral Nutrition 2022, 46, 12–41.” (page 19, line 438-448)
Comment 2-9: …it should be noted that, given that this is an observational study, they should be rephrased in terms of association, that is, “was associated with” rather than “contributes to.”
Response: As your comment, we corrected the phase "contributes to" have been revised to "was associated with".
Edited the manuscript are as follows
“This study provides novel evidence that adequate PN supplementation exceeding 1000 kcal/day was associated with faster recovery and shorter hospitalization in geriatric patients with infectious colitis.” (page 13, line 283)
Comment 2-10: As limitations, the possibility of reproducibility in another center and the need for future prospective studies should be indicated.
Response: We fully agree with the reviewer’s comment regarding external validity and the need for further verification. As a single-center retrospective study, our findings may reflect specific local practices or patient demographics.
We have explicitly added a statement in the Limitations section acknowledging that reproducibility in other centers has not yet been established and emphasizing the necessity of future prospective, multicenter studies to validate our results and confirm the causal relationship between caloric provision and clinical outcomes.
Edited the manuscript are as follows
“Furthermore, the reproducibility of these findings in other healthcare settings remains to be established due to the single-center design. Additionally, by focusing exclusively on short-term hospitalization outcomes, the study was unable to assess longer-term endpoints such as readmission, post-discharge mortality, or nutritional recovery. These limitations highlight the need for future multicenter, prospective studies with larger sample sizes, extended follow-up, and more comprehensive evaluation of potential confounders to validate and expand upon these findings.” (page 12-13, line 275-281)
